# A Study of the Adoption and Implementation of Enterprise Resource Planning (ERP): Identification of Moderators and Mediator

**Md. Aftab Uddin [1],\*** , **Mohammad Sarwar Alam [2],\*** , **Abdullah Al Mamun [1],**
**Tohid-Uz-Zaman Khan [3] and Ayesha Akter [1]**

1    Department of Human Resource Management, University of Chittagong, Chattogram 4331, Bangladesh;
     mamun09@cu.ac.bd (A.A.M.); ayeshakter@cu.ac.bd (A.A.)
2    Department of Management, University of Chittagong, Chattogram 4331, Bangladesh
3    Department of Accounting, University of Chittagong, Chattogram 4331, Bangladesh; tohidkhan@cu.ac.bd
*    Correspondence: mdaftabuddin@cu.ac.bd (M.A.U.); alam.sarwar@cu.ac.bd (M.S.A.)

**Abstract:** Given the dearth of studies in developing and Asian countries' context, the present study attempts to excavate the predictors of enterprise resource planning (ERP) adoption and implementation. Based on the Unified Theory of Acceptance and Use of Technology (UTAUT) model and open innovation literature, an extended model is proposed encompassing mediator and moderator variables. The study follows the deducting reasoning approach with the positivism paradigm. Out of 235 responses, the study used 225 replies collected through a self-administered sampling, and the data were analyzed by using PLS-based structural equation modeling. The study revealed that the hypothesized direct influences are significant except the influence of facilitating conditions on actual use. Likewise, the intention to use mediates the impact of facilitating conditions on the actual use of ERP. However, there is no moderating effect of education and firms' size among the hypothesized influence. The study contributes to advance the previous findings by using an extended UTAUT model and validates results with the rest of the world.

**Keywords:** intention to use ERP; actual use of ERP; enterprise resource planning; open innovation; UTAUT

## 1. Introduction

The past century has witnessed a robust development of dynamic firms with the attribution of the globalization, the fourth industrial revolution, and technological advancement [1]. As a result, firms of the 21st century are windswept to open innovation that enables the whole world turning into a global village [2–5]. Recently, technological breakthrough turns into a sine qua non for the survival and resilience of any firm [6–8]. As businesses are confronting for faster, better and cheaper than ever, technology-based open innovation linking inbound and outbound partners has become a crucial demand for any firm to reap competitive advantage, and to turn burgeoning challenges into possibilities [2,9,10]. Thereby, ensuring open innovation through the usage of enterprise resource planning (ERP) has turned out to be a prime mover of building dynamic capabilities for realizing a firm's sustainable competitive advantage [11,12].

In open innovation contexts, an organization needs to inter-connect with multiple users in order to generate any novel solutions [13]. The ERP supports an organization to keep pace with open innovation systems by keeping the outsider (suppliers and customers) in and the insider (internal processes, and employees) out through technological networking [13–16]. Henceforth, a deeper insight into the ERP adoption and implementation encloses all the functional core processes to deal with all

the value-creating entities in upstream and downstream in an open innovation context [2,14,17,18]. The ERP system integrates numerous core processes through connecting the data and material flow and renders functionality, features and capabilities [19,20]. It optimizes firms' performance through a complete centralization of data and information flow across multiple functional areas by eliminating redundant data and information [21–23].

Apart from technical, operational, and strategical benefits, a quick preview of prior research spectacles a wide variety of benefits in a collaborative manner, such as acceleration of performance, productivity, streamlining business processes, efficiency, customer service and effectiveness, and the reduction of cost and energy, bottleneck in management communication, wastage of materials and cycle times [22,24–26]. However, many factors are presumed to exist for approval of ERP adoption and implementation, such as financial capabilities of the firm, availability of specialists and infrastructures, among others [19,27]. Globally, it is integrated module-based software that provides a wholesome network of knowledge and material flows among collaborative processes of any firm along with its upstream and downstream end-users [28]. The usage of ERP gives cutting-edge competitive advantage through automated collaborative networking by the elimination of duplicated activities across various functional areas [22,29,30].

Extant pieces of literature asserted exponential growth of ERP adoption and implementation in developing countries' contexts because of its significant contributions to make work-process faster, better, and cheaper [2,31]. Nonetheless, no debate arises concerning the acceptability of ERP for business process transformation, the applications of this open innovation technology in developing nations' context, particularly in Asian case, is meager [22,32–34]. Regrettably, only 7% of firms adopt ERP as their business process solution [35]. Common issues in the slow proliferation of ERP is addressed in numerous studies. Alhirz [36], Rajapakse and Seddon [35], Rajan and Baral [37] and Alam and Uddin [2] attested that one of the main reason for developing countries' context, specifically in Asia, is the absence of behavioral intention to actual use than technological literacy on technical specifications. Studies also posited that lack of technical, financial and organizational supports prevents the prevalence of ERP elsewhere.

Moreover, prior studies also asserted that technical sophistication and frequent change in the module specifications are retarding the mushrooming growth of ERP adoption and implementation [34]. Therefore, the complexities due to the collaborative intents involved in ERP adoption and implementation attribute to a grand failure in many contexts [38,39]. Henceforth, we propose the following research questions:

RQ1:   What are the dominant factors influencing ERP adoption and implementation?
RQ2:   Which are the driving mediator and moderators triggering the adoption and implementation of ERP in a developing country's context?

The present study contributes to advance the previous findings in numerous ways. First, we documented numerous global studies, which are seemed to Western-biased, to unearth the factors triggering the reluctance of end-users, and reasons for gigantic failure from yielding massive success [2,40]. Surprisingly, very few studies were observed in the developing countries' contexts to explore the reasons regarding its slow progress [32,37]. As mentioned in prior studies, the studied results validate the previous findings explored in various settings in the rest of the world in general [11,19,37], and an Asian case in particular. Second, prior studies documented conflicting results on using facilitating conditions [41–44]. Unlike the findings of Chao [44] and Dwivedi, Rana, Jeyaraj, Clement and Williams [43], results in the application of the original Unified Theory of Acceptance and Use of Technology (UTAUT) model yield insignificant direct effects on intention to use, and actual use. To abate the conflicting findings, we, thus, further examined the influence of facilitating conditions in both direct and indirect mechanisms with the help of mediating and moderating effects to validate the previous findings in a different context. Finally, as per our knowledge go, we do not observe any study using intention to use as a mediator between facilitation conditions and actual use of ERP, and education and

firms' size as moderators between intention to use and actual use of ERP. To advance the premise of open innovation, we extended the previous UTAUT model by the inclusion of mediator and moderators. In a way, the moderating effects of ERP users' education and firms' size and mediating impact of the behavioral intention of the influence of facilitating conditions on the actual use will add new evidence to extend and advance the previous findings.

The remaining part of the study follows the inclusion of theoretical background and formulation of the hypotheses based on extensive literature reviews. Subsequently, the study outlines the methods section and reports on observed findings. The discussion section illustrates the results in light of previous empirical and theoretical underpinnings. Finally, we conclude the study signifying managerial implications for the practicing managers and ERP users along with the directions for future research basing on limitations that prevent the generalizability and causal inference for the present findings.

## 2. The Theoretical Background of the Study

Recently, the burgeoning growth of new technological developments has caught attention from academics and professionals in the management science and engineering discipline [2]. In the last two decades, existing literature observed that there is an increasing trend of studies on new technology adoption to catch up with the tenet of open innovation [45]. Hence, ERP adoption and implementation deal with how individuals accept and use new technology [2,43]. Shreds of evidence and linkages have explained the adoption and implementation of technology from numerous theories, such as the theory of planned behavior, the innovation diffusion theory, the technology acceptance model, the social cognitive theory, and UTAUT [43,44,46].

In the last decade, the technology acceptance model, among other theories, was widely used for the adoption and usage of any technology [41,42]. Additionally, numerous papers were advocated with the addition or elimination or combination of theories into an integrated model for exploring their similarities and dissimilarities [47]. Henceforth, prior studies observed serious reservations while using those fragmented theories because critics demonstrated negative sanctioning on those disjointed theories that failed to portray a holistic view of understanding of individual behavior during adoption and the actual use of technology [48]. Furthermore, many theories explaining the adoption and implementation of technology, particularly ERP, in a closely held innovation, ignoring the outside in and inside out. Besides, numerous theories attributing to a single conceptual model yield little sense of it [43,47].

The present study has used a dominant view of the adoption and implementation of ERP through the usage of UTAUT in an open innovation context [49,50], via linking an outsider in and an insider out [45]. Despite earlier studies on adoption and implementation of technology used a number of theories such as the theory of planned behavior, the theory of reason action, technology acceptance model, diffusion of innovation, and learning theories, etc., they are seemed, now, that those have less capability in explaining the intention and the actual use of ERP [49,51]. Keeping abreast of those changes with technology, buyers, and service providers in a boundary-less open innovation context, Venkatesh, Morris, Davis and Davis [41] and Venkatesh et al. [52] advocated that an integrated theory replacing those fragmented theories (previously used eight key theories), which could better explain how human and society impact adoption and implementation of technology [44,47,53,54]. Following the tenet of open innovation theory, we included one mediator and two moderators with the widely used UTAUT to ensure the robustness of the findings in the observed context.

The supremacy of using these theories underlies in its consideration of multi-level perspectives [51]. The study of Mahmood, Uddin and Luo [6] and Uddin et al. [55] signify that multi-level variables are to be considered in open innovation cases. The UTAUT, based on a comprehensive review and synthesis of several theoretical models [2,41,53], constitutes five distinct constructs in multi-level perspectives, such as performance expectancy and effort expectancy (individual-level variables), social influences (social level variable) and facilitating conditions (organizational level variable), and behavioral intention for predicting actual use behavior (individual-level variables) [41,42].

## 3. Hypothesis Development

### 3.1. Direct Effects

Performance expectancy reveals users' perception of the ability to ensure its functional capability to result in a specific solution or behavior [41]. Venkatesh, Morris, Davis and Davis [41] illustrated that "the magnitude to which an end-user believes that the use of the given application program will assist in arriving at a particular solution or job performance". It verifies the users perceived belief of the performance capability toward a new technology [56]. The utility, job-technology fit, and perceived benefits of a given ERP withstand the end-users' perceived intimidation of potential negative consequences that prevent the intention to use ERP [57]. In line with the discussion, empirical findings in various contexts also confirmed that performance expectancy is a vital influencer of users' intention to adopt ERP [2,37]. Thus, it is hypothesized as follows:

**Hypothesis 1.** *Performance expectancy predicts the intention to use an ERP.*

Effort expectancy measures how ease the use of technology [53]. According to the tenant of UTAUT, it is asserted that users incline to adopt a technology if the later serves their purpose [53]. Similar to the findings of performance expectancy, studies for the adoption of any technology showed that effort expectancy is also a stronger predictor of intention to use a technology [41–44]. In a situation when end-users of ERP perceive that the ease of using the system, it facilitates their inclination to use it [58]. In a way, Alam and Uddin [2] and Rajan and Baral [37] approved that effort expectancy results in the intention to use an ERP in developing countries' context. We also observe similar findings with studies in various contexts, such as e-learning [53], m-banking [59–61], ICT [54], internet use [51], animation and story-telling [56], etc. Henceforth, users will not use any technology if they believe that the perceived ease of use is in the bracket [53,62]. The following hypothesis is developed:

**Hypothesis 1.** *Effort expectancy influences users' intention to use an ERP.*

While taking the decision concerning any person's social phenomenon, individuals care essential others, such as individuals in his/her society regarding the latter evaluation of the former behavior [44]. The social influence can be stated as the magnitude to which an individual believes and perceives how his/her essential others expect him to use or not to use a new system [41]. It is also observed that people are influenced by peers, relatives, friends, and essential others during their decision-making [41,63]. In both UTAUT and revised UTAUT, academics revealed that the intents of a person are shaped by how their essential others (relatives, colleagues, friends and neighbors) expect them [60,64]. Expectations, evaluations, and normative beliefs from their important neighborhood regulate individuals to behave in a certain way regarding what to buy and choose [65,66]. Thus, the existing literature posits that essential others' pressure and expectation will have a significant influence on the adoption of ERP [43,44,47,67]. Likewise, the studies of Alam and Uddin [2] attested that the intention to behave is significantly influenced by what important others expect from the self. The following hypothesis is proposed:

**Hypothesis 3.** *Social influence influences users' behavioral intention to adopt ERP.*

To adopt new technology, it is prerequisites to exist in the organizational and technical infrastructure of any technology. Studies showed mixed findings on the influence of facilitating conditions. Whereas studies excerpted that facilitating conditions impacts behavioral intention to use a technology [2,43,61], it is also documented that other streams of studies reported no significant influence of facilitating conditions on the intention to use a technology [47,48,66]. Arguably, from the essence of UTAUT, we posit that the availability of facilitating conditions accelerates intention to use of technology, particularly ERP [2]. Thus, facilitating conditions, such as perceived compatibility, and technical and infrastructural supports must support the use of a new system [41]. Before making any decision to

adopt and buy any technology, as per the premise of UTAUT supports people look for the availability of technical specifications and other infrastructural supports because the absence of them stimulates their ambiguity or negligence for any future emergencies [52,68]. Importantly, an organization lacking resources, knowledge, and supports from the management would likely procrastinate the adoption and implementation of ERP [2]. Perceived supports from the organization of sanctioning required resources give rise to the possibility of adoption and implementation of ERP [56]. In a way, users' intention to adopt and finally use ERP would rise if they can be assured of the availability of facilitating conditions in their firms [43,44,47]. In line with the understanding of UTAUT and previous empirical findings, the following hypotheses are developed:

**Hypothesis 4.** *Facilitating conditions influences the intention to use ERP.*

**Hypothesis 5.** *Facilitating conditions affects the actual use of ERP.*

Yu [69] asserted that individual usage behavior is predictable in the domain of the psychological discipline. Henceforth, in the field of management science and engineering, the impact of individual behavioral intention on actual use is widely studied [42]. The actual use refers to the manifestation of an observable response in a predictable context concerning a given target [70]. We observed no major disagreement on the usage of the intention to use ERP for the actual use of ERP. Rather, many studies evidenced that behavior intention to use is the only predictor, if not the only, of the actual use of any technology. On the other hand, few studies concerning the application of UTAUT demonstrate that intention to behave is one of the vital factors defining the actual use of a technology [54,65]. Moreover, studies of Alam and Uddin [2], Carlsson, et al. [71] and Rajan and Baral [37] demonstrated that intention to use ERP influences the actual use of ERP significantly in their studies. Thus, a documented summary of the existing studies leads to hypothesize in the following manner:

**Hypothesis 6.** *Intention to use ERP predicts the actual use of ERP.*

*3.2. Mediating and Moderating Effects*

In the existing literature, there is a little disagreement on the influence of facilitating conditions of behavioral intention to use and actual usage. Following the axiom of UTAUT, we can easily relate the influence of facilitating conditions as a direct, and an indirect predictor of actual use [42]. It is, nonetheless, also evident that facilitating conditions is a direct predictor of actual use [47,53]. Gripping of these debates, in H4, we hypothesize that facilitation conditions predicts intention to use ERP basing on both empirical and theoretical underpinning [37,52,68]. We also approve that the intention to use ERP significantly influences the actual use of ERP based on prior studies [2,37]. In line with documented findings of Alam and Uddin [2], Dwivedi, Rana, Jeyaraj, Clement and Williams [43], Raza, Shah and Ali [61], we can advance with theoretical and empirical observations that the impact of facilitating conditions on the intention to use ERP results in the actual use of ERP. Thus, the following hypothesis is developed:

**Hypothesis 7.** *Intention to use ERP mediates the influence of facilitating conditions on the actual use of ERP.*

Users' education may influence their opportunities, preferences, and choices later in their life. The relationship between education and the use of any technology may operate in any formal and informal settings [72,73]. Accordingly, the education of ERP users might influence their choices to use and adoption of technology [74]. The study revealed that highly educated employees tend to adopt new technologies more quickly than those who are less educated [75] because of the costs and uncertainty associated with the adoption of new technology reduced by proper education and the flow of information. The positive regression and correlations between education and technology use

and adoption hypothesized and proposed in previous studies [72–74]. Thereby, the level of education (higher/lower) might result in the extent of the influence of intention to use ERP on the actual use of ERP.

**Hypothesis 8.** *ERP users with an advance educational profile will have a stronger influence on the positive impact of intention to use ERP on the actual use of ERP than with a less advanced educational profile.*

Our preview of earlier studies on information technology adoption suggested that organizational size is one of the strongest predictors of it [76]. Surprisingly, in the case of the UTAUT model, we have noticed very few studies with organizational size as a factor to testify whether it matters in the adoption and implementation of technology. Importantly, large enterprises have the stronger capability of managing risk, abundant available resources, and robust infrastructure to adopt technologies than small and medium enterprises [76,77]. Askarany and Smith [78] and Ko, et al. [79] argued that factors that are needed to adopt new technologies are widely available in large firms than the small one. On the contrary, significant problems of any small organization are they suffer from the lack of resources, financial difficulties, and scarcity of professionals; these things lead to challenges in adopting the technology [79,80]. Thereby, the moderating influence of a firm's size is hypothesized as follows:

**Hypothesis 9.** *The positive influence of intention to use ERP on the actual use of ERP is influenced more in a large firm than in a small firm.*

Henceforth, the research model based on the UTAUT as well as the empirical findings is displayed in Figure 1. Figure 1 demonstrates that the present study adds new pieces of evidence and extends the original UTAUT by adding a direct line from facilitating conditions toward actual use in order to determine if any mediation mechanism exists in between them. The theoretical model also encloses two moderator variables, i.e., the size of the firm and education literacy of the users of the impact of intention to use ERP on the actual use behavior of ERP based on the ideation of the studies of Affes and Ayadi [81]; Askarany and Smith [78]; Lleras-Muney and Lichtenberg [75]; and Wozniak [82]. Figure 1 mirrors the gauging of the users' adoption and implementation of ERP in a multi-level perspective as individual behavior is influenced by the presence of individual, group, organizational level variables.

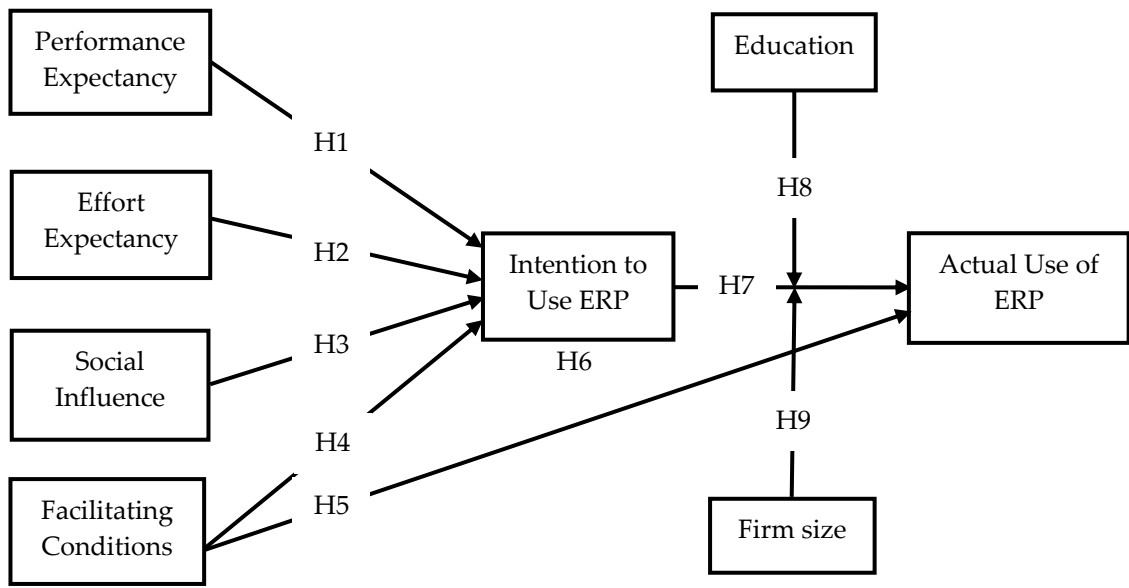

**Figure 1.** Conceptual framework of the study.

## 4. Research Methodology

### 4.1. Research Design

The research design applied deductive reasoning approach to examine hypothesized relationships. The multi-item scales that measures constructs were adapted from prior studies to collect data from ERP users during the period from January 2019 to March 2019. We sent a survey questionnaire to the Human Resource Division of the concerning organizations whose employees are using ERP. To prompt their easy and comfortable response, a self-addressed envelope was sent to them while delivering them survey-questionnaire [83]. PLS-based structural equation modeling (PLS-SEM) is used to analyze the data, which is popularly used in management science in the last four decades. Scholars recommended the PLS-based SEM model over simple regression because it guarantees the robustness of the result via investigating the entire model rather than testing each path alone [83–85].

### 4.2. Data Collection Procedure

The present research was applied in manufacturing organizations, where ERP is extensively used. Survey questionnaires were delivered among employees at different levels in diverse organizations. It guarantees the robustness of the findings from the general representation of samples across organizations [2]. 235 replies were received through the self-administered survey from 410 distributed questionnaires in local and multinational organizations, which yields a 58.75 percent response rate, which is adequate to apply the PLS-path model [86] and much higher than similar studies in different contexts [18,26,87]. The self-administered survey was executed because it saves time and cost with a maximum response rate [2]. 225 replies were used finally, leaving ten responses with missing and unmatched answers.

### 4.3. Measurement Tools

Measurement tools from prior studies were utilized, which were used in different parts of the world. A few changes were made to the items underlying the construct for ensuring their face validity to represent accurate meaning among the respondents. Constructs underlying performance expectancy, effort expectancy, social influence, facilitating conditions, and intention to use are measured using the constructs developed and refined by Venkatesh, Morris, Davis and Davis [41] and Venkatesh, Thong and Xu [42], and actual use is measured using the items refined by Rajan and Baral [37].

### 4.4. Bias Concern

Method bias and response bias are two significant concerns preventing the authenticity of the research outcomes [88,89]. Hence precautionary measures were taken in advance to guard the negativity of biasness issues [55]. Initially, the respondents were assured of their anonymity which driven them to provide us accurate responses without any phobia of identity disclosure [84,86,90]. Furthermore, it is asserted that the present study would report on general industrial phenomena instead of highlighting on any specific firm's work-processes which incites them truly reflect their provoking thoughts of their daily working environment. Additionally, we experimented Harman's one-factor test to explore if there is any single factor explaining more than 50 percent of the total variance [84]. Estimates sanctioned that not a single factor explains more than 50 percent of the total variances. Finally, the correlation matrix is also examined to see if there is any association of more than 80 percent between two variables [91,92]. Notably, results revealed that the highest correlation between any two variables is 0.622. Thus, there is no issue response bias observed in this study [93,94].

## 5. Results

### 5.1. Sample Characteristics

Employees' gender, age, education, tenure, rank, and the size of the firms were modeled in the study as control variables. We coded demographic variables with distinct values, such as gender

(1 = male, 2 = female), age (1 = 18–25 years old, 2 = 26–30 years old, and 3 = more than 30 years old), education (1 = Master, and 2 = others), tenure (1 = 1–4 years old, 2 = 5–9 years, and 3 = 10 years and more), and the size of the firm (1 = small and medium, and 2 = large organizations). Demographic data revealed that workplaces are still male-dominated (male = 64 percent and female = 36 percent) and the average age is 29 years old, and 26 to 29 years old dominates the age group enclosing in this study. Besides, of the total respondents, respondents with experience of 1 to 5 years are the major participants (109 with a percentage of 48.44) in this study. Finally, respondents from large organizations represent the largest segment (148 of 225 respondents) of the respondents.

## 5.2. Model Evaluation

The PLS-led structural equation modeling (PLS-SEM) was used to test the proposed hypotheses. Both the measurement model and structural model are two vital aspects of any SEM, which are evaluated by testing reliabilities, validities, cross-loading, beta-coefficient, coefficient of determination, and path-significance [85,95]. Bootstrapping with sample cases of 5000 was used to test the results.

## 5.3. Measurement Model Evaluation

The measurement model is tested through the assessment of reliability scores and validity scores. We also checked the confirmatory factor analysis using cross-loading tables. Table 1 displays the reliability and validity scores of this study. Reliabilities showed that both Cronbach's alpha and composite reliabilities are above the minimum threshold limit of 0.70 [6,84,94]. Estimates showed the minimum Cronbach's alpha and composite reliability scores are 0.821 (performance expectancy), and 0.837 (effort expectancy), respectively. Thus, no adverse concern is found in reliability scores.

**Table 1.** Reliability, convergent, and discriminant validities test reports.

| Control Variables | 1 | 2 | 3 | 4 | 5 | 6 | 7 | 8 | 9 | 10 | 11 |
|---|---|---|---|---|---|---|---|---|---|---|---|
| 1. Age | 1 | | | | | | | | | | |
| 2. Experience | 0.843 ** | 1 | | | | | | | | | |
| 3. Education | 0.198 ** | 0.123 | 1 | | | | | | | | |
| 4. Firm Size | 0.326 ** | 0.126 | 0.555 ** | 1 | | | | | | | |
| 5. Gender | −0.287 ** | −0.205 ** | 0.062 | 0.091 | 1 | | | | | | |
| **Latent Variables** | | | | | | | | | | | |
| 6. PE | 0.007 | 0.037 | −0.031 | −0.038 | −0.031 | 0.900 | | | | | |
| 7. EE | −0.042 | 0.036 | −0.025 | 0.053 | 0.013 | 0.596 ** | 0.921 | | | | |
| 8. SI | 0.057 | 0.048 | 0.021 | 0.065 | −0.052 | 0.622 ** | 0.531 ** | 0.915 | | | |
| 9. FC | 0.012 | 0.048 | 0.000 | 0.006 | 0.021 | 0.531 ** | 0.436 ** | 0.452 ** | 0.887 | | |
| 10. IU | 0.081 | 0.078 | −0.026 | 0.049 | −0.067 | 0.592 ** | 0.515 ** | 0.567 ** | 0.489 ** | 0.940 | |
| 11. AU | −0.064 | −0.088 | −0.130 * | 0.150 ** | 0.015 | 0.373 ** | 0.436 ** | 0.348 ** | 0.364 ** | 0.605 ** | 0.921 |
| Mean | 29.03 | 3.59 | - | - | - | 3.733 | 3.646 | 3.710 | 3.672 | 3.873 | 3.833 |
| SD | 5.055 | 3.310 | - | - | - | 0.862 | 0.975 | 0.866 | 0.870 | 0.897 | 0.885 |
| CA | - | - | - | - | - | 0.821 | 0.841 | 0.835 | 0.909 | 0.934 | 0.911 |
| CR | - | - | - | - | - | 0.844 | 0.837 | 0.854 | 0.936 | 0.938 | 0.944 |
| AVE | - | - | - | - | - | 0.809 | 0.849 | 0.837 | 0.786 | 0.883 | 0.849 |

PE. Performance expectancy, EE. Effort expectancy, SI. Social influence, FC. Facilitating conditions, IU. Intention to use, AA. Actual use, SD. Standard deviation, CA. Cronbach's alpha, CR. Composite reliability, and AVE. The average variance extracted. **,*: Correlation is significant at the 0.01 and 0.05 level respectively.

Validities were tested in terms of convergent validity and discriminant validity. Convergent validity examined to observe if the items representing a variable are accurately converged to the underlying variable or not. The average variance extracted (AVE) is estimated and assessed whether any construct's AVE score is less than 0.50. Our tested score showed that the minimum AVE of any construct is 0.786, which is above the minimum threshold limit (AVE > 0.50). An AVE score of more than 0.50 signifies that no issue of convergent validity is present in this study.

In order to examine the discriminant validities, the square root of the AVE of all constructs was evaluated. The result in Table 1 exhibited that the square root of the AVE of all constructs is higher than their correlations with other constructs. It strengthens the notion that all the constructs are

discriminately valid. The CFA is examined using the cross-loading [96]. An examination of Table 2 (cross-loading) demonstrates that all the items representing a variable were rather highly loaded to their constructs. Estimates on the cross-loading table showed that no item is packed highly other than its underlying construct. Thus, no issue is observed about the discriminant validity of any constructs [96]. Therefore, it can be opined that the measurement model evaluation demonstrates a good fit in terms of reliability and validity [84,86].

**Table 2.** Confirmatory factor analysis using cross-loading.

| Items | BI | EE | FC | PE | SI | AU |
|-------|------|------|------|------|------|------|
| iu1 | 0.950 | | | | | |
| iu2 | 0.922 | | | | | |
| iu3 | 0.946 | | | | | |
| ee1 | | 0.919 | | | | |
| ee2 | | 0.918 | | | | |
| ee3 | | 0.936 | | | | |
| ee4 | | 0.913 | | | | |
| fc1 | | | 0.884 | | | |
| fc2 | | | 0.896 | | | |
| fc3 | | | 0.879 | | | |
| fc4 | | | 0.888 | | | |
| pe1 | | | | 0.898 | | |
| pe2 | | | | 0.906 | | |
| pe3 | | | | 0.898 | | |
| pe4 | | | | 0.897 | | |
| si1 | | | | | 0.922 | |
| si2 | | | | | 0.896 | |
| si3 | | | | | 0.924 | |
| si4 | | | | | 0.918 | |
| au1 | | | | | | 0.939 |
| au2 | | | | | | 0.905 |
| au3 | | | | | | 0.920 |

*5.4. Structural Model Evaluation*

Evaluation of the structural model is tested using β, *p*-value, and $R^2$. In Figure 2, observation of βs and *p*-values of path relations showed that all the paths excepting facilitating conditions to the actual use of ERP are found significant with a *p*-value of less than 0.029 (<0.05). Coefficient of determination ($R^2$) asserted that estimates are candidly better since minimum $R^2$ is 0.372 (the actual use of ERP) [83,95]. Furthermore, we studied the effect size of the impact of exogenous variables on their endogenous variables [6,94].

Cohen [97] and Wetzels, et al. [98] mentioned that the minimum effect size, which is mentioned as the goodness of fit (GoF) in Equation (1), is to be 0.10, 0.25, and 0.30 for a small, medium, and significant effects on endogenous variable respectively, provided that minimum AVE must be higher than 0.50 [99]. The employed GoF, as directed by Tenenhaus, et al. [100] and Wetzels, Odekerken-Schröder and Van Oppen [98], is estimated as the square root of the times of mean AVE (average variance extracted) and mean $R^2$. The calculated result, in Equation (1), showed that GoF is 0.588, with a minimum AVE of 0.786. Thus, the effect size is large (>0.30) with all the preconditions are fulfilled [97–100].

$$\text{GoF} = \sqrt{\text{Average AVE} \times \text{Average } R^2} \tag{1}$$

$$\text{GoF} = \sqrt{0.414 \times 0.836}$$

$$\text{GoF} = 0.588$$

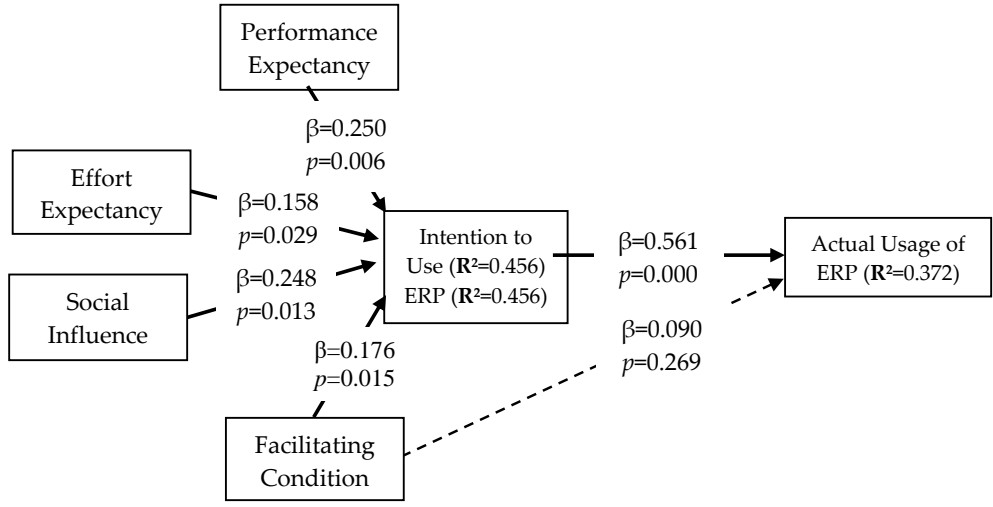

**Figure 2.** The PLS model along with path estimates.

## 5.5. Testing Direct Effects

Table 3 demonstrated that all the hypotheses (H1: $\beta = 0.25$; $p = 0.007$, H2: $\beta = 0.158$; $p = 0.037$; H3: $\beta = 0.248$; $p = 0.004$, H4: $\beta = 0.176$; $p = 0.021$, and H6: $\beta = 0.561$; $p = 0.000$) are found significant, excepting H5 ($\beta = 0.091$; $p = 0.289$). Estimates revealed that H5 is insignificant because of their dominant indirect effects from facilitating conditions to intention to use ERP to the actual use of ERP. Thus, the present findings support all the hypotheses relating to direct effects, and only H5 is not supported.

**Table 3.** Testing hypotheses with direct effects.

| Hypothesis | Path Relations | B | Standard Error | T-Statistics | $p$-Values | $R^2$ | Decision |
|---|---|---|---|---|---|---|---|
| H1 | PE → IU | 0.250 | 0.092 | 2.714 | 0.007 | | Supported |
| H2 | EE → IU | 0.158 | 0.076 | 2.093 | 0.037 | 0.456 | Supported |
| H3 | SI → IU | 0.248 | 0.086 | 2.873 | 0.004 | | Supported |
| H4 | FC → IU | 0.176 | 0.076 | 2.323 | 0.021 | | Supported |
| H5 | FC → AU | 0.090 | 0.084 | 1.061 | 0.289 | 0.372 | Not supported |
| H6 | IU → AU | 0.561 | 0.088 | 6.409 | 0.000 | | Supported |

## 5.6. Testing the Mediating Effect

To have a mediation effect, two conditions are to be observed. First, the independent variable must have a significant direct effect (c) on the dependent variable before adding a mediating variable [101,102]. Second, the significant direct effect (c) before running the mediation effect must disappear (for full mediation) or reduce (for partial mediation) after using the mediating variable in the previous unmediated relationship [6,83,84,86,102]. Table 4 exhibits significant direct effects ($\beta_c = 0.368$, $p = 0.000$) before running mediator variables. Despite both the indirect paths are significant ($\beta_a = 0.176$, $p = 0.021$; $\beta_b = 0.561$, $p = 0.000$), the direct path ($c'$) after using mediating variable turned into insignificant ($\beta_{c'} = 0.090$, $p = 0.289$). In a way, there is a full mediation of intention to use of the impact of facilitating conditions on the actual use of ERP. Thus, H7 is supported.

**Table 4.** Mediating effect estimates.

| | Path Relations | Beta Coefficient | Standard Error | T-Statistics | $p$-Value | Indirect Effect | Total Effect | Sobel Test |
|---|---|---|---|---|---|---|---|---|
| H7 | FC → AU (c) | 0.368 | 0.064 | 5.718 | 0.000 | | | |
| | FC → IU (a) | 0.176 | 0.076 | 2.323 | 0.021 | 0.099 | 0.189 | Z = 3.307 |
| | IU → AU (b) | 0.561 | 0.088 | 6.409 | 0.000 | | | $p < 0.00$ |
| | FC → AU (c') | 0.090 | 0.084 | 1.061 | 0.289 [ns] | | | |

FC. Facilitative conditions, IU. Intention to use, AC. Actual use, Indirect effect = $a * b$, total effect = $c'$ + indirect effect, ns. Not significant.

### 5.7. Testing Moderating Effects

The moderating effects of the education level of the respondents and firms' size are examined. Using SmartPLS2, the moderating effects are displayed in Figure 3. In H8, the moderating effect of respondents' education on the influence of intention to use ERP on the actual use of ERP is investigated. The interaction effect revealed that there is no significant moderating influence ($\beta = -0.002$; $p = 0.980$) of education on the hypothesized influence. Therefore, H8 is not supported.

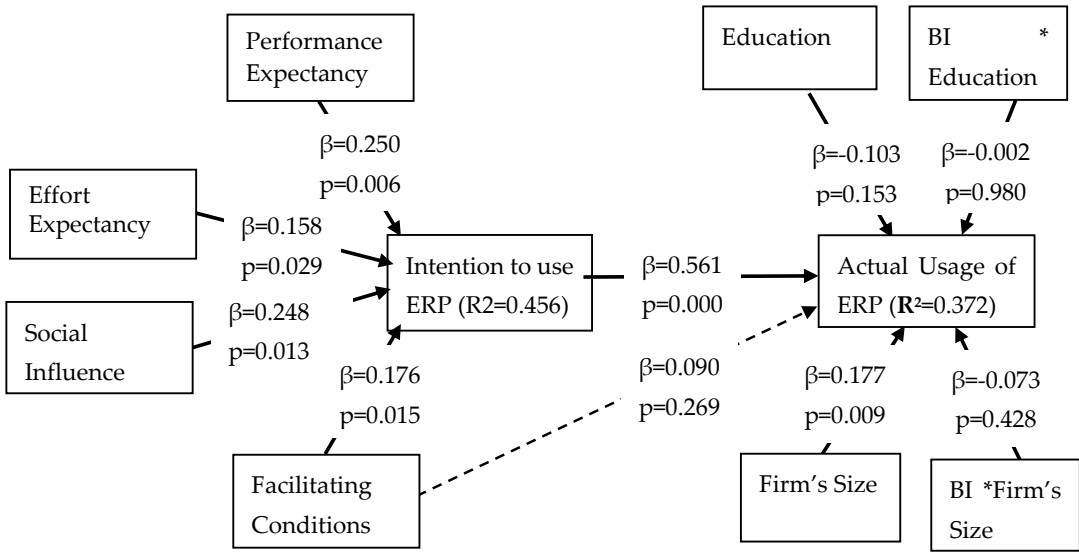

**Figure 3.** Moderating effects of education and firms' size in the structural model.

Finally, the moderating influence of firms' size (H9) of the hypothesized influence of intention to use on the actual use of ERP is also tested. Similarly, the interaction effect of firms size demonstrated no significant influence ($\beta = -0.073$; $p = 0.428$). It approves that there is no moderating effect of firms' size on the previously hypothesized influence of intention to use ERP on the actual use of ERP. Accordingly, H9 is also not supported. The previous estimates are plotted in the diagram in Figure 4a,b. Figure 4a manifested that a higher level and lower level of education does not neutralize its negative relations between intention to use ERP and the actual use of it. Likewise, Figure 4b also delineated that larger firms' size and smaller firms' size do not posit any negative but significant moderating influence on the hypothesized relationship.

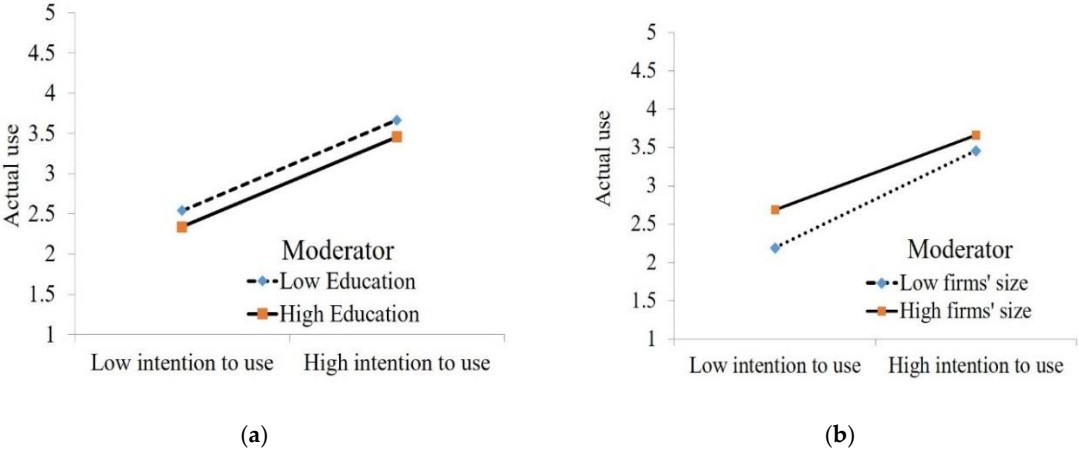

**Figure 4.** (**a**) Moderating effect of Education. (**b**) Moderating effect of Firms' size.

## 6. Discussion and Conclusion

Surviving in the era of technological sophistication and information superhighway is challenging [30,45]. Open innovation brings a phenomenal change by connecting myriads of organizational operational areas with outsiders, such as suppliers and customers, and insiders, such as employees and management [1,16,103]. To support an organization's open innovation, adoption and implementation of technology, such as ERP makes it possible because it brings knowledge and material inflows and outflows together. In this study, a conceptual research model is developed, underlying the extended UTAUT model along with mediating and moderating mechanisms, attempting to unearth the predictors of ERP adoption and the actual use of it.

In line with our endeavored model, we searched the literature to map the areas for contributing and advancing knowledge and conducted a self-administered survey on the actual users of ERP. The results illustrated that performance expectancy, effort expectancy, social influence, and facilitating conditions are significantly influencing intention to use ERP while the influence of facilitating conditions on the actual use of ERP is not supported. In our mediation analysis, we detected that full mediation exists. However, none of the moderating influences is found significant. This study definitely will contribute to advance the literature of the UTAUT model in an open innovation context and validate previous findings in other contexts. Essentially, policy-makers, government agencies, academicians and industry professionals engender insightful understanding regarding the mechanism of ERP adoption and the actual use of it.

The study examined factors influencing the adoption and implementation of ERP in a developing country's context, particularly in a South Asian Nation's context using the essence and understanding of the UTAUT model. Tables 3 and 4, and Figures 3 and 4a,b demonstrated the revealed findings. In line with the tenet of UTAUT [42,52], the results showed that all the direct influencers predict their outcome variables, leaving facilitation conditions toward the actual use of ERP insignificant. It is observed that similar findings were found with the rest of the world, particularly with ERP adoption and implementation in the study of Rajan and Baral [37], Awa, Uko and Ukoha [19], Dwivedi, Rana, Jeyaraj, Clement and Williams [43] and Alam and Uddin [2]. The study supported the hypotheses indicates that when performance expectancy, effort expectancy, social influence, and facilitating conditions of ERP prevail, users' passion and inclination toward behavioral intention to use of ERP grows, and vice versa [32]. Unlike the findings of Salloum and Shaalan [53], Chao [44], and Dwivedi, Rana, Jeyaraj, Clement and Williams [43], the influence of facilitating conditions is not found influential statistically. Surprisingly, the insignificant impact is also evidential in the study of Hoque and Sorwar [104] regarding actual technology used when the users depend on externalities to affirm their choices.

We further tested direct and indirect effects of facilitating conditions toward the actual use of ERP. The premise of ERP adoption and implementation is built on UTAUT. Interestingly, prior studies failed to investigate if the mediation effect of intention to use ERP works or not on the hypothesized relationship between facilitating conditions and the actual use of ERP [2,37,41,42]. Ingenuously, they conducted study unpretentiously assuming that full mediation exists. Keeping the naivety of previous studies, and to fill the gap; thereby, we attended the research and found that full mediation exists since direct effect from facilitating conditions to the actual use of ERP is insignificant. It happens because an indirect effect through intention to use ERP bridges facilitating conditions and the actual use of ERP is so stronger that the direct influence of facilitating conditions on the actual use of ERP seems unnecessary.

A perusal of the moderating effect of firms' size and education level of users, who are associated with ERP, exposed that neither the impact of education nor the effect of firms' size significantly moderates hypothesized influences. The results are not consistent with the thoughts of earlier studies [72–74]. To our belief, the reason for the insignificant moderating influence of education is that the acute emergence of ERP acceptance and use of it is indifferent to the magnitude of the ERP users' education level. Surprisingly, the moderating influence of firms' size is also not conferred with the existing pieces of literature [79,80]. Moreover, it is also believed that the solutions provided by the module-based

application programs of ERP software are highly regarded irrespective of the firms' size. Indirectly, it coincides with the provoking thoughts that "size does not matter for the intention to use and actual use of ERP." Surprisingly, it is to be noted that the direct effect of organizational size is positively linked, meaning that firms' size positively influences the usage of ERP adoption.

### 6.1. Managerial Implications of the Study

In the age of sharp competition, continuous innovation is a must to reap competitive advantage. Firms with open innovation in a collaborative effort guarantee the former to withstand any competition. In line with the conceptualization of open innovation theory, ERP integrates internal processes and eliminates all the redundancies along with keeping outsiders such as suppliers and customers connected with upstream and downstream material and information flows. Henceforth, the study extends and advances the previous UTAUT by including additional moderators and mediator to explore the actors responsible to adopt and implement ERP in a developing country context, particularly in an Asian hierarchical society, Bangladesh. In this quest, the study proposes practical implications for managers, professionals, and organizations. The outcomes of this study enclose perception for ERP users and managers to execute ERP adoption and implementation. The findings of this study will also facilitate businesses and entrepreneurs to understand which factors are influential and where to invest in reshaping behavioral intention to use ERP in an open innovation context. Additionally, IT professionals, policymakers and ERP vendors will be highly benefited by exposing the insights drawn from the present findings. In a way, the study stimulates ERP vendors to produce ERP application programs uniquely designed for need-based requirements of the market and end-users.

Additionally, unlike the findings of prior studies [34,42,44,105], the present study explores the potential impact of education and organizational size. Despite it shows that the emergence of ERP usage is at each kind of organizations irrespective of their size and employees' education level, the result, however, posits positive direct influence of organizational size on actual use of ERP. It outlines that the large size of the organization can afford ERP adoption and implementation more than small size organizations. Unlike the findings of Barrane, Karuranga and Poulin [47], Dwivedi, Rana, Jeyaraj, Clement and Williams [43], and Chao [44], the current study explores that actual use of ERP is ameliorated by the indirect effect of intention to use ERP than the direct influence of facilitating conditions. Thus, the policy-maker and ERP vendors should emphasize on facilitating conditions catering to stimulating behavioral intention to use.

### 6.2. Limitations of the Study and Future Research Directions

There is no disagreement on the merit of the study, and contributions to advance and extend the previously held knowledge. However, the study is also not above limitations. Data collection procedure poses several constraints, such average age of the respondents is 29 years old, and nearly 50 percent of the respondents have tenure experience of 5 years. Thus, the studied result will expose to partial results. Mainly, the respondents are too young to decide on ERP-related issues. Hence, a diverse workforce with age and tenure differentials might impede critics on the ground of tenure or age bias. Another noteworthy aspect is its sample size, which is only 225. We suppose that the sample size is quite a few to decide on this strategic issue regarding the behavioral aspect of technology adoption [86]. Accordingly, the results, in the Bangladesh context, are statistically significant, more studies with the increasing breadth of scope and larger sample size might strengthen the robustness of the model and generalizability of the findings [37]. Similar to the results of Costa, Ferreira, Bento and Aparicio [34], the present study collected data from several organizations, which lacks the comprehensiveness of industry-wide panorama. Notably, the present study did not consider the primary moderators in the original UTAUT model, such as age, gender, experience, and voluntariness; because we saw that, the users are young and relatively fresh employees who have been using ERP. However, it inhibits us from accurate generalization on the causal influence without applying the original UTAUT model in its entirety. The nature of the data, such as cross-sectional data, makes the results less generalizable and

valid. Hence, future researchers might conduct a study using longitudinal and repeated-longitudinal data in order to validate and generalize the findings in other similar/dissimilar contexts. The study has been executed in a developing country, particularly in Bangladesh, which reflects a country bias. Thus, the future researchers might conduct their study in similar or dissimilar contexts or cross-cultural study to observe the generalizability of the current and other findings. Finally, this study tests the moderating influences of education and the firm's size, and the influence of other moderators is entirely ignored. Therefore, it is also recommended for future researchers to investigate the influence of other confounding factors such as age, gender, experience, and voluntariness for examining their potential interaction effects.

**Author Contributions:** Conceptualization, M.A.U., M.S.A., A.A.M., T.-U.-Z.K. and A.A.; methodology, M.A.U.; software, M.A.U.; validation, M.S.A., A.A.M., T.-U.-Z.K. and A.A.; formal analysis, M.A.U.; investigation, M.S.A.; resources, A.A.M., T.-U.-Z.K. and A.A.; data curation, M.S.A.; writing—original draft preparation, M.S.A., A.A.M., T.-U.-Z.K. and A.A.; writing—review and editing, M.A.U.; visualization, A.A.M., T.-U.-Z.K. and A.A.; supervision, M.A.U.; project administration, M.A.U. and M.S.A.; funding acquisition, A.A.M., T.-U.-Z.K. and A.A. All authors have read and agreed to the published version of the manuscript.

**Funding:** This research received no external funding.

**Acknowledgments:** We are very thankful to Rupam Kumar Das, Anupam Kumar Das, and Rubina Easmin for their cooperation and supports during our manuscript preparations.

**Conflicts of Interest:** The authors declare no conflict of interest.

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
