# Peer review of "A Study of the Adoption and Implementation of Enterprise Resource Planning (ERP): Identification of Moderators and Mediator"

_2199-8531, doi:10.3390/joitmc6010002_

Round 1

Reviewer 1 Report

This study reports on „A study of the adoption and implementation of enterprise resource planning (ERP): Identification of moderators and mediator”. I am pleased that the authors have chosen this topic. The written style, logic structure, the introduction, and methodology provide a useful explanation for the results or reach a solid conclusion.

I ask that the authors specifically address each of my comments in their response. I hope my comments, observations, and suggestions will allow the authors to improve the manuscript and work towards publication. Below, I include comments pointing toward some of the issues.

Originality

Research contribution in the paper can be identified; this study can be justified as innovative.

Title

The title is correct as it reflects the objective of the work.

Abstract

The abstract provides a structured summary including contextual background, and result.

The theoretical background of the study and  the Hypothesis development

These sections are very well based. The theoretical considerations in the reviewed study were based on a large number of current literature items. Approximately 105 literature sources are presented; all references are accurate and correct. There is plenty of updated and recent literature in the reference list, and data sources are updated as well.

Research methodology

The authors did a tremendous work. The concept of the model is great and the calculations are mathematically correct. The problem with it is that using the model name is wrong. They did not estimate the SEM model, but the PLS-PATH model. Please correct this mistake.

Discussion and Conclusion

The Discussion and Conclusion sections are very well based. I found the Managerial implications of the study and the Limitations of the study and Future research directions. Congratulations.

On the basis of these observations, this paper is recommended for publication after the revisions detailed in the research methodology part.

Author Response

Response to Reviewer 1:

Respected Reviewer

Thanks for your useful comments and constructive suggestions for the revision. As per your suggestion, we revised/corrected the paper.

Comment 1: The problem with it is that using the model name is wrong. They did not estimate the SEM model, but the PLS-PATH model. Please correct this mistake.

Our response: We revised as advised by the reviewer.

Thank you so much for your well-thoughts appreciation on the originality, title, abstract, theoretical background and hypothesis development, and discussion and conclusion.

Also thank you very much for your recommendation to publish after these additions.

Thank you so much.

Regards,

Authors.

Reviewer 2 Report

The authors made significant corrections in the content and the presentation of the manuscript which has been significantly improved.

Author Response

Response to Reviewer 2:

Respected Reviewer,

Thanks to reviewer (2) for valuable time and encouraging comments. We appreciate your comments to improve the merit of the paper.

Comment 1: The authors made significant corrections in the content and the presentation of the manuscript which has been significantly improved.

Our response: Thank you so much.

Considering the merit, we do hope that it is acceptable now.

Thank you so much

Authors

This manuscript is a resubmission of an earlier submission. The following is a list of the peer review reports and author responses from that submission.

Round 1

Reviewer 1 Report

Overall, the concept of this paper is interesting although it suffers from several limitations (both structural and content-wise) and it can be significantly improved proofreading and re-organizing several axes.

First of all, Introduction does not provide any information about the purpose of the paper and the structure of the paper. Furthermore, the practical implications of this survey should be expanded. The contribution of this paper, the research gap as well as the motivation should be mentioned in order to justify the value of this paper.

The paper does not demonstrate an adequate understanding of the relevant literature in the field. The author(s) should discuss the results of previous studies related to the UTAUT theory and the adoption of ERP systems in order to support the conceptual model. The conceptual model should present the hypotheses of this paper. Author(s) should analyze the findings and research gaps from previous researchers. Furthermore, more references should be added in order to refine better the theoretical foundation of this paper and also provide more research insights.

The author(s) did not discuss much if at all the methodology behind analyzing data. More information about what they actually did should be provided. How the responders were selected? Furthermore, the values of the following indexes should be added: CFI, GFI, NFI, SB x2, df, RMR, RMSEA, p-close.

Finally, the current version of the manuscript may not show strong research motivation/contribution. The findings are a good basis for discussion but they need more conceptualization to make the contribution of the research more evident. The author(s) may need to clearly document, justify, and clarify the practical and theoretical implications of their study as well as limitations and suggestions for future research. Author(s) should answer the following questions in order to make the contribution of the paper explicit in conclusion:         

What does this research tell us that we didn’t already know? What is the contribution of the most significant results of the paper? What are the limitations of this paper and suggestions for future research?

Author Response

Response to Reviewer 3:

Respected Reviewer

Thank you so much for your comments to improve the merit of the paper. We made revision according to your feedbacks:

Comment 1: This is a Path modeling method, it contains all the methodological elements you need and everything is done correctly. Authors have to make it clear that they did not perform a CB-SEM (covariance-based structural equation model) but a PLS-PATH model (Partial Least Squares Path) or PLS-SEM. This is not obvious in the article. Please provide the effect size and power estimations for the main two regressions.

Our response: We responded it and showed effect size through GoF estimation and demonstrated in structural model.

Comment 2: English language and style: moderate English changes required. 

Our response: We took helps to correct errors in academic writing.

Reviewer 2 Report

This paper endevaours to determine influential factors of ERP adoption and implementation by applying the well-known structural equation modelling technique.

I found the topic interesting and the consideration of firm size as influential factor is surprisingly really a novelty in the field. The used English is clear and sophisticated almost in the entire text, which is also one of the merits of the paper.

However, I suggest some important modifications before the publication of the paper.

First, at least one methodological alternative of the Structural Equation Modelling should be mentioned in the paper in the topic of attribute significance questionnaire based surveys, such as the multi-criteria decision making methods. I recommend to cite the model of Duleba (Duleba, S. An AHP-ISM approach for considering public preferences in a public transport development decision. Transport, 1-10. https://doi.org/10.3846/transport.2019.9080), in which the importance and the influence of the attributes were simultaneously analysed. The applied methodlogy should be compared to this model and proved why SEM is the better technique for the ERP case or why it was selected.

Further, there are some minor errors that should definitely be corrected before publication.

line 204: which yields

line 316-320: it is not readable, please correct

line 441-442: the two sentences have the same meaning

line 443: 50 percent of the respondents

Having corrected the above mentioned issues, I recommend the paper for publication.

Author Response

Response to Reviewer 2:

Respected Reviewer,

Thanks to reviewer (2) for valuable time and encouraging comments. We appreciate your comments to improve the merit of the paper. As per reviewer’s suggestion, we revised/corrected the paper.

Comment 1: First, at least one methodological alternative of the Structural Equation Modelling should be mentioned in the paper in the topic of attribute significance questionnaire based surveys, such as the multi-criteria decision making methods. I recommend to cite the model of Duleba (Duleba, S. An AHP-ISM approach for considering public preferences in a public transport development decision. Transport, 1-10.

Our response: We revised methodological part in accordance with the reviewer’s comment. Also, we cited the recommended paper at their relevant texts.

Comment 2: Further, there are some minor errors that should definitely be corrected before publication; line 204: which yields; line 316-320: it is not readable, please correct; line 441-442: the two sentences have the same meaning; line 443: 50 percent of the respondents.

Our response: We addressed these issues at their relevant texts.

Thank you so much

Authors

Reviewer 3 Report

This study reports on „A study of the adoption and implementation of enterprise resource planning (ERP): Identification of moderators and mediator”. I am pleased that the authors have chosen this topic. This is an interesting and timely focus. Given the dearth of studies in developing and Asian countries’ context, the present study attempts to excavate the predictors of enterprise resource planning (ERP) adoption and implementation. The written style has a logic structure. The work is original.

I ask that the authors specifically address each of my comments in their response. I hope my comments, observations, and suggestions will allow the authors to improve the manuscript and work towards publication. Below, I include comments pointing toward some of the issues.

Originality

Research contribution in the paper can be identified; this study can be justified as innovative.

Title

The title is correct as it reflects the objective of the work.

Abstract

The abstract provides a structured summary including contextual background, and result, conclusion, and implications of key findings, etc.

The theoretical background of the study

The most relevant part is theoretical background that gives a perfect context for the justification of the research. This section includes many relevant references and authors provide solid theoretical foundations for the analysis using appropriate references. Objectives are clearly stated and achieved.

Research method

This is a Path modeling method, it contains all the methodological elements you need and everything is done correctly. Authors have to make it clear that they did not perform a CB-SEM (covariance-based structural equation model) but a PLS-PATH model (Partial Least Squares Path) or PLS-SEM. This is not obvious in the article. Please provide the effect size and power estimations for the main two regressions.

Results

The tables presented here are justified for an adequate statement of results.

Discussion and Conclusion

Conclusions are clear and follow logically form the text.

English language and style: moderate English changes required. 

On the basis of these observations, this paper is recommended for publication.

Author Response

Response to Reviewer 3:

Dear Reviewer

Thank you so much for your comments to improve the merit of the paper. We made revision according to your feedbacks:

Comment 1: This is a Path modeling method, it contains all the methodological elements you need and everything is done correctly. Authors have to make it clear that they did not perform a CB-SEM (covariance-based structural equation model) but a PLS-PATH model (Partial Least Squares Path) or PLS-SEM. This is not obvious in the article. Please provide the effect size and power estimations for the main two regressions.

Our response: We responded it and showed effect size through GoF estimation and demonstrated in structural model.

Comment 2: English language and style: moderate English changes required. 

Our response: We took helps to correct errors in academic writing.

Please feel free to contact us for any further concern regarding the paper.

We look forward to hearing from you.

Best regards,

The authors

Round 2

Reviewer 1 Report

The authors made significant corrections in the content and the presentation of the manuscript which has been significantly improved. However, they can make more corrections in several axes in order to increase the contribution of the paper.

First of all, in Introduction the authors added information about the purpose of the paper and the structure of the paper. Furthermore, they mentioned the practical implications of this survey the contribution of this paper, the research gap as well as the motivation. However, the first and second paragraphs of the Introduction should focus on the need for the ERP adoption and the benefits for businesses. Furthermore, a paragraph about the use of ERP adoption in Asia should be added.

The paper does not demonstrate an adequate understanding of the relevant literature in the field. The author(s) should discuss the results of previous studies related to UTAUT theory and the adoption of ERP systems in order to support the conceptual model. Author(s) should analyze the findings and research gaps from previous researchers. Furthermore, more references should be added in order to refine better the theoretical foundation of this paper and also provide more research insights.

The hypotheses of this paper should be justified by a discussion of the variables that present. For example, the 1st hypothesis includes two variables (performance expectancy and intention to use ERP), so authors should discuss results from previous studies regarding these two variables. Furthermore, the positive or negative relationship between the variables in each hypothesis should be mentioned. Otherwise, the support or rejection of the hypotheses cannot be evaluated.

The author(s) did not discuss much if at all the methodology behind analyzing data. More information about what they actually did should be provided. How the responders were selected? Is the sample adequate? Which was the sample of previous surveys? Section 4.2 should be referred in the 5th section.

The 5th section should be renamed as Results. According to CFA, factors that have loadings <0.5 should be removed from the model. Please revise the factor analysis.

The findings are a good basis for discussion but they need more conceptualization to make the contribution of the research more evident. Please add a conclusion before the 7th section and merge these two sections.

Finally, references were updated but the new references are not aligned with the purpose of the paper. Authors should add more recent references regarding ERP adoption and UTAUT model and particularly in Asia. Furthermore, they could add the following references regarding the strategic use of Information Systems and the effects on performance:

Kitsios, F. and Kamariotou, M. (2019). Strategizing Information Systems: An empirical analysis of IT Alignment and Success in SMEs, Computers, 8 (4), 74, pp. 1-14 Kitsios, F. and Kamariotou, M. (2019). Information Systems Strategy and Strategy-as-Practice: Planning Evaluation in SMEs, Proceedings of Americas Conference on Information Systems (AMCIS2019), Cancun, Mexico, pp. 1-10 Kamariotou, M. and Kitsios, F. (2019). Strategic Planning and Information Systems Success: Evaluation in Greek SMEs, Proceedings of the  21st IEEE Conference on Business Informatics (CBI2019), Moscow, Russia, pp. 204-211

Author Response

Response to Reviewer 1:

Respected Reviewer

Thank you, respected reviewer, so much for your valuable time and encouraging comments. We appreciate your comments to improve the merit of the paper. As per your suggestions, we revised/corrected manuscript, which is marked in green in the main text.

Comment 1: the first and second paragraphs of the Introduction should focus on the need for the ERP adoption and the benefits for businesses. Furthermore, a paragraph about the use of ERP adoption in Asia should be added

Our response: We introduction sections as per the reviewer’s guideline.

Comment 2: The paper does not demonstrate an adequate understanding of the relevant literature in the field. The author(s) should discuss the results of previous studies related to the UTAUT theory and the adoption of ERP systems in order to support the conceptual model. Author(s) should analyze the findings and research gaps from previous researchers. Furthermore, more references should be added in order to refine better the theoretical foundation of this paper and also provide more research insights.

Our response: We revised the contributions part in 2nd last paragraph, theoretical part and also updated references from recent papers.

Comment 3: The hypotheses of this paper should be justified by a discussion of the variables that present. For example, the 1st hypothesis includes two variables (performance expectancy and intention to use ERP), so authors should discuss results from previous studies regarding these two variables. Furthermore, the positive or negative relationship between the variables in each hypothesis should be mentioned. Otherwise, the support or rejection of the hypotheses cannot be evaluated.

Our response: We improved the hypotheses section with debates with more recent papers.

Comment 4: The author(s) did not discuss much if at all the methodology behind analyzing data. More information about what they actually did should be provided. How the responders were selected? Is the sample adequate? Which was the sample of previous surveys?

Our response: We revised methodology section and responded all the questions.

Comment 4: Section 4.2 should be referred in the 5th section.

Our response:  Section 4.2 is transferred to section 5.

Comment 5: The 5th section should be renamed as Results. According to CFA, factors that have loadings <0.5 should be removed from the model. Please revise the factor analysis.

Our response: We revised the text file. We also eliminated loaded factors in other constructs, particularly to deal with those less than 0.50

Comment 6: Please add a conclusion before the 7th section and merge these two sections.

Our response: We revised manuscript accordingly.

Comment 7. Authors should add more recent references regarding ERP adoption and UTAUT model and particularly in Asia. Furthermore, they could add the following references regarding the strategic use of Information Systems and the effects on performance:

Kitsios, F. and Kamariotou, M. (2019). Strategizing Information Systems: An empirical analysis of IT Alignment and Success in SMEs, Computers, 8 (4), 74, pp. 1-14

Kitsios, F. and Kamariotou, M. (2019). Information Systems Strategy and Strategy-as-Practice: Planning Evaluation in SMEs, Proceedings of Americas Conference on Information Systems (AMCIS2019), Cancun, Mexico, pp. 1-10

Kamariotou, M. and Kitsios, F. (2019). Strategic Planning and Information Systems Success: Evaluation in Greek SMEs, Proceedings of the  21st IEEE Conference on Business Informatics (CBI2019), Moscow, Russia, pp. 204-211 

Our response: We added more references regarding ERP and UTAUT along with the three papers mentioned by reviewers.

Thank you so much

Authors

Round 3

Reviewer 1 Report

The authors made significant corrections in the content and the presentation of the manuscript which has been significantly improved. However, they could make some corrections to improve the content of the theoretical background and the conclusion section.

The authors could expand the discussion about H1, H2, H6, and H7. They can add more information about the results of previous surveys regarding the variables that are included in these hypotheses. In conclusion, the authors have provided a discussion about the limitations of this paper. They can suggest for future researchers to conduct the same survey in other countries or in other industries and compare the results. Also, they can use other variables to test this model (for example age, gender, experience, and voluntariness). Finally, authors could use passive voice through the text and avoid using active voice and refer to them.

Author Response

Response to Reviewer 1:

Respected Reviewer

Thank you so much for your comments to improve the merit of the paper. We made revision according to your feedbacks:

Comment 1: The authors could expand the discussion about H1, H2, H6, and H7. They can add more information about the results of previous surveys regarding the variables that are included in these hypotheses.

Our response: We revised the relevant hypotheses.

Comment 2: In conclusion, the authors have provided a discussion about the limitations of this paper. They can suggest for future researchers to conduct the same survey in other countries or in other industries and compare the results. Also, they can use other variables to test this model (for example age, gender, experience, and voluntariness).

Our response: We revised limitations and included the reviewer’s suggestion in the directions for future research.

Comment 3: Finally, authors could use passive voice through the text and avoid using active voice and refer to them.

Our response: We used passive form and minimized the active voice.

Thank you so much

Best regards,

The authors
